# The prognostic value of gait speed in hemodialysis patients: A prospective observational study

Joyce Noelly Vitor Santos[1,2,3], Vanessa Gomes Brandão Rodrigues[2,3], Redha Taiar[4], Tamara Cunha[3], Elisângela Andrade Assis Madeira[1,3], Inara Caroline Marcelino Martins[1], Maria Cecília Sales Mendes Prates[1], Vanessa Kelly da Silva Lage[1,2,3], Ana Cristina Rodrigues Lacerda [1,2,3], Henrique Silveira Costa[1,3], Frederico Lopes Alves[2], Emílio Henrique Barroso Maciel[1], Pedro Henrique Scheidt Figueiredo[1,3], Vanessa Amaral Mendonça [1,2,3]*

1 Programa de Pós-graduação em Reabilitação e Desempenho Funcional, Universidade Federal dos Vales do Jequitinhonha e Mucuri, Diamantina, Minas Gerais, Brazil, 2 Programa de Pós-graduação em Ciências da Saúde, Universidade Federal dos Vales do Jequitinhonha e Mucuri, Diamantina, Minas Gerais, Brazil, 3 Laboratório de Inflamação e Metabolismo (LIM), Universidade Federal dos Vales do Jequitinhonha e Mucuri – UFVJM, Diamantina, Minas Gerais, Brazil, 4 MATIM, Université de Reims Champagne Ardenne, Reims, France

* vaafisio@hotmail.com

## Abstract

### Background

Gait speed has emerged as a sensitive and practical measure of functional status, and its association with overall health and adverse outcomes in various populations has been increasingly recognized. However, its prognostic value among hemodialysis patients remains insufficiently explored.

### Objective

To assess the prognostic value of the gait speed in hemodialysis patients.

### Methods

This prospective observational study assessed adults with end-stage renal disease on hemodialysis. Baseline measurements were taken from April 2019 onward, and survival was observed until December 2024. Usual walking speed was measured by an 8-meter gait speed protocol, with the time to cover the central 4-meter distance recorded. Also, established prognostic factors were evaluated. Data were analyzed using the Cox regression analysis and the ROC curve. The Kaplan-Meier curve compared the cumulative survival across gait speed categories. Statistical significance was set at 5%.

**Data availability statement:** All relevant data are within the manuscript and its Supporting Information files.

**Funding:** The authors thank the Coordenação de Aperfeiçoamento de Pessoal de Nível Superior (CAPES), Brazil; Conselho Nacional de Desenvolvimento Científico e Tecnológico (CNPq) (409837/2024-5; 303706/2024-4 to V. A.M.), Brazil; Fundação de Amparo à Pesquisa do Estado de Minas Gerais (FAPEMIG) (APQ-00709-24; APQ-01522-23), Brazil; and Hemodialysis Center of Santa Casa de Caridade de Diamantina, Minas Gerais, Brazil.

**Competing interests:** The all authors have declared that no competing interests exist.

## Results

A total of 120 eligible patients were included. Age (HR 1.05 [95% CI 1.01–1.09], p = 0.01), phosphorus (HR 0.72 [95% CI 0.54–0.96], p = 0.03), and gait speed (HR 0.04 [95% CI 0.01–0.14], p < 0.0001) were associated with mortality. Gait speed (HR 0.04 [95% CI 0.01–0.15], p < 0.0001) was an independent predictor of mortality. The optimal cutoff point for mortality risk identification was ≥ 1.13 m/s (AUC = 0.85 [0.77–0.0.90]; p < 0.0001), and low mobility was associated with a 10.4-fold higher mortality risk (HR 10.49 [95% CI 3.70–29.96], p < 0.0001).

## Conclusion

Gait speed presented excellent accuracy in mortality risk identification, and low mobility was a significant risk factor for death in individuals undergoing hemodialysis. These findings highlight that gait speed assessment, a simple, quick, and low-cost measure, can be implemented in dialysis centers for risk stratification, supporting more targeted clinical decisions and therapeutic approaches. Furthermore, they underscore the importance of functional assessment in this setting.

## Introduction

The prevalence of end-stage kidney disease (ESKD) is a relevant problem worldwide [1,2] and has increased substantially due to the increasing incidence of risk factors such as hypertension and diabetes [3]. Despite advances in dialysis treatments, this population remains highly vulnerable, with elevated rates of morbidity and mortality [4–6].

In patients with ESKD undergoing hemodialysis, the high symptom burden [7] and the complex interaction of the pathological mechanisms, such as mineral disorders [8], progressive bone disease [8], cachexia [9], chronic inflammation [10], physical inactivity [11], and environmental factors, result in the impairment of overall health, including reduced functional capacity [11,12] and increased fragility [13].

The functional capacity has emerged as a determinant marker of prognosis in various chronic conditions. [14,15] In this regard, the gait speed is a valid, reliable, and sensitive measure appropriate for assessing and monitoring functional status and overall health in various populations. In older people, the gait speed is designated as the "6th vital sign" [16] and the "functional vital sign" [17]. In hemodialysis patients, studies have shown that gait speed is slower based on general population values [18,19] and is associated with the diagnosis of frailty [20], hospitalizations [5], and cardiovascular events [21,22]. Recent studies have also demonstrated an association between gait speed and mortality, [5,21,23] identifying it as a predictor of all-cause mortality in developed countries [21,23]. However, its prognostic value among hemodialysis patients remains insufficiently explored.

Thus, this study investigated the hypothesis that gait speed may be an accurate tool for mortality risk identification and that patients with low mobility, measured for

the gait speed test, may be at a higher risk of death. Accordingly, this study aimed to assess the prognostic value of gait speed in hemodialysis patients.

## Patients and methods

### Study design

This was a prospective observational study conducted in a reference Brazilian hemodialysis center. This study adhered to the ethical principles of the Declaration of Helsinki, approved by the institutional ethics committee (CAAE: 60169822.1.0000.5108), and all participants provided written informed consent prior to their inclusion in the study. Our research strategy followed the Recommendations of PROGRESS (Prognosis Research Strategy) [24]. The STROBE statement was used as a guide to report this study [25,26].

### Participants

Patients on hemodialysis treatment older than 18 years who were receiving hemodialysis treatment thrice weekly for at least three months were included in the study. Exclusion criteria were pregnant women, significant mental illness, contra-indications, or inability to perform the functional tests, patients who underwent transplants or who changed dialysis centers and lost contact during follow-up.

### Data collection

The recruitment period started on 09/04/2019 and ended on 31/12/2024. Baseline assessments included anthropometric measurements (weight, height, and body mass index) and usual walking speed. These evaluations were conducted before hemodialysis during the second weekly session. Subsequently, body composition measurements, including the appendicular lean mass (ALM) and body fat, were performed after hemodialysis using Dual-Energy X-ray Absorptiometry (DEXA). The sector itself carried out the biochemical analysis.

### Usual walking speed

Patients were instructed to walk a distance of eight meters at their usual gait speed, and the time taken to cover the middle four meters was recorded. The patient was permitted to use a walking aid device during the test. The usual walking speed was measured twice for the four-meter gait speed, and the shortest time recorded was selected [27,28].

### Known factors evaluated

Demographic variables (sex and age), anthropometric measurements, clinical factors (presence of diabetes mellitus and hemodialysis vintage), and biochemist variables (hemoglobin, phosphorus [P], iron [Fe], Alkaline phosphatase) were evaluated due to their recognized prognostic value in hemodialysis patients. The methods used to perform the biochemical analyses were briefly described in the supplementary material (S1).

### Follow-up period and outcome measures

The follow-up was started immediately after the baseline assessments and was carried out through weekly visits by a researcher to the dialysis center. The endpoint was defined as death, irrespective of the cause. After follow-up, the sample was allocated into mortality and survival groups.

### Sample size calculation

A priori sample size calculation was performed to ensure that the monitored sample adequately addressed the study question. The OpenEpi software was used for the sample size calculation, applying an alpha error of 5%, a statistical power of 95%, and a ratio of events of 34%, the minimum sample size required was determined to be 84 patients.

## Data analysis

Statistical analysis was performed using SPSS version 22.0 (SPSS Inc., Chicago, IL, USA) and MedCalc Statistical Software version 13.1 (MedCalc Software, Ostend, Belgium). The data distribution was verified using the Kolmogorov–Smirnov test. Continuous variables were shown as mean and standard deviation (normal distribution) or median and interquartile range (non-normal distribution). Group comparisons were performed through independent t-tests and Mann–Whitney's or Chi-Square tests.

Univariate Cox regression analysis evaluated the association between gait speed and mortality. If $p < 0.5$, multivariate analysis was performed to assess the hazard ratio of the gait speed to predict mortality after adjusting for age, sex, hemoglobin, P, Fe, Alkaline phosphatase, and presence of diabetes.

A Receiver-operating Characteristic (ROC) analysis was performed to determine the sensitivity and specificity of different cutoff values of gait speed and predict mortality events. The area under the ROC curve (AUC) and 95% confidence interval (CI) were calculated for all tests, and optimal cutoffs were determined by values with the best combination of sensitivity and specificity using the Youden Index. An AUC greater than 0.7 was considered acceptable, while an AUC greater than 0.8 was considered excellent for the proposed cutoffs [29,30]. Alternative cutoff points were suggested considering the combination of specificity and sensitivity. The Positive Predictive Value (PPV) and Negative Predictive Value (NPV) were calculated. The Kaplan-Meier curve was performed to evaluate the survival distribution equality for the different categories of gait speed from the established cutoff. Statistical significance was set at 5%.

## Results

### Characteristics of subjects

One hundred and twenty subjects were eligible for participation and enrolled in the study (Fig 1). At baseline, participants exhibited an average age of 51.37 ± 15.45 years, with a dialysis vintage of 3.22 (3.75) years. Among these, 76 (63%) were male, and 44 (47%) were female. The mean follow-up time was 31.37 ± 21.54 months. Mortality occurred in 23% of the sample. The causes of death included acute respiratory failure (25%), sepsis (12.5%), sudden death (18.7%), cancer (12.5%), stroke (6.2%), hemorrhagic shock (6.2%), and cases of undetermined origin (18.7%).

Characteristics of the participants are presented in Table 1. Age and serum phosphorus levels were significantly higher in the mortality group. Gait speed was significantly lower in the mortality group. No differences were observed in sex, dialysis time, diabetes presence, body composition, and other biochemical variables.

### Cox regression analysis between gait speed and mortality

In model 1, while age, phosphorus, and gait speed were significantly associated with mortality in univariate analysis, in the adjusted model, only gait speed was significantly associated with mortality. In model 2, low mobility was independently associated with mortality (Table 2).

### ROC curves for gait speed

The area under the ROC curve (AUC) to identify mortality hazards in hemodialysis patients by gait speed was 0.85 (0.77–0.90), $p < 0.0001$. The gait speed was excellent for screening mortality (Fig 2). Table 3 shows the properties of the cutoff points with the best combination of sensitivity and specificity and negative and positive predictive values of gait speed tests for screening mortality.

### Kaplan-Meier curve analysis

Accumulative survivals were significantly decreased in hemodialysis patients with gait speed ≤1.13 or >1.13 m/s (Log-rank: $x^2 = 34.38$; $p < 0.0001$) (Fig 3).

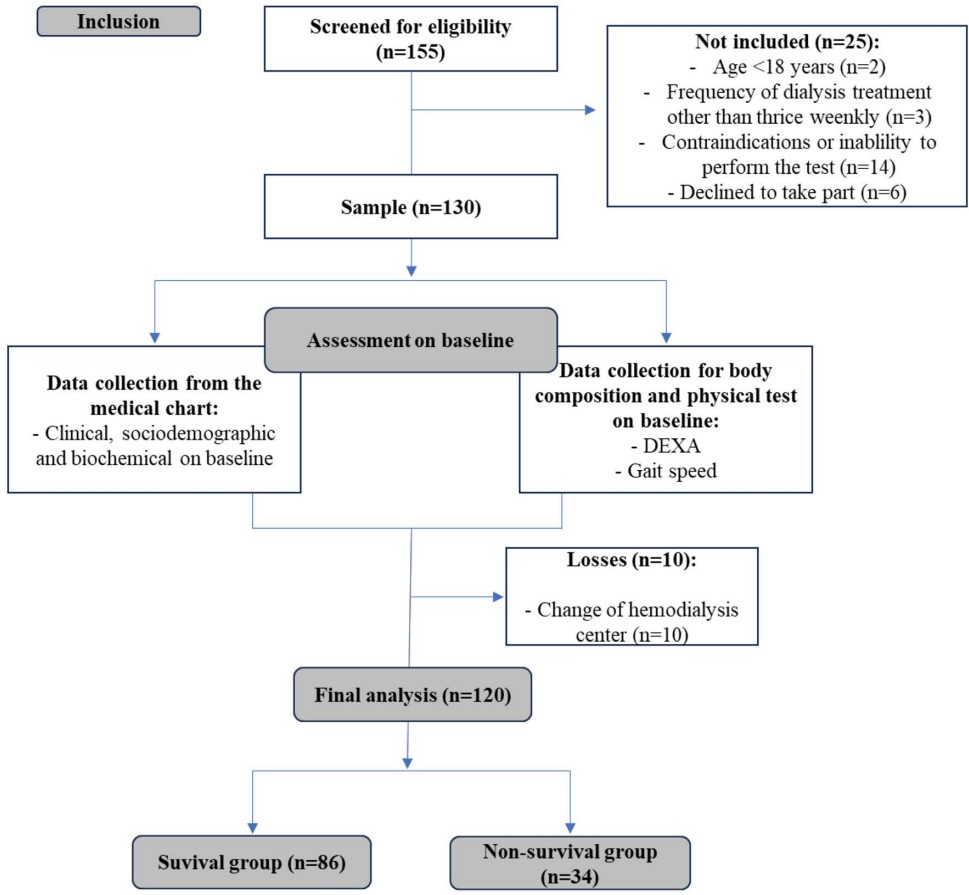

**Fig1. Flowchart of participant selection and monitoring.** DEXA, Dual-Energy X-ray Absorptiometry.

## Discussion

The present study showed the prognostic value of the usual gait speed in hemodialysis patients. The main findings of this study were: (1) the gait speed is an independent predictor of mortality, (2) patients with low mobility, measured by gait speed, had a 10.4-fold higher risk of death, and (3) knowledge of the predictive value of gait speed ranges. These results have significant clinical meaning since they demonstrated that a simple and inexpensive evaluation can be valuable in clinical assessment and risk stratification of patients, which can assist in early therapeutic management.

This study showed that non-survivors were older, had had higher levels of albumin and lower levels of phosphorus, and had lower mobility values than survivors. Serum phosphorus levels differed between the survival and non-survival groups, both groups presented mean values within a relatively preserved range (survival: 5.04 ± 1.51 mg/dL; non-survival: 4.31 ± 1.40 mg/dL). Hyperphosphatemia is a defined prognostic factor in hemodialysis patients, however previous studies indicate that increased mortality risk is particularly associated with low phosphorus levels (e.g., < 3 mg/dL) [31,32] often reflecting malnutrition or protein-energy wasting. In this context, lower phosphorus concentrations in our sample may act as a surrogate marker of poorer nutritional and functional status rather than indicating a direct causal relationship.

Additionally, the gait speed was our sample's independent predictor of death. In other populations, the gait speed was associated with mortality, such as individuals with older age, [33] chronic obstructive pulmonary disease, [34,35] and

**Table 1. Comparison of the sample demographics, biochemists, body composition, and gait speed between mortality and survival groups.** Data presented as mean±SD or median (interquartile range). Kt/v fractional urea clearance, PTH parathormone, C-RP C-reactive protein, LDL low-density lipoprotein, HDL high-density lipoprotein, BMI body mass index, ALM appendicular lean mass *p<0.05, #gait speed <1.13 m/s.

| Variables | All (n=120) | Survival (n=86) | Non-survival (n=34) | p-value |
|---|---|---|---|---|
| Sex (M/F) | 76 (63)/ 44 (37) | 56 (65)/ 30 (35) | 20 (59)/ 14 (41) | 0.76 |
| Age (years) | 51.37±15.45 | 47.60±15.57 | 60.90±10.29 | <0.0001* |
| Dialysis vintage (years) | 3.22 (3.75) | 2.04 (3.54) | 4.17 (4.08) | 0.07 |
| Diabetes (yes n%) | 32 (27) | 20 (23) | 12 (35) | 0.29 |
| **Biochemist** | | | | |
| Vit D (ng/ml) | 37.50 (18.00) | 37.40 (14.00) | 29.60 (25.00) | 0.06 |
| PTH (pg/ml) | 341.35 (314.00) | 308.60 (377.00) | 307.50 (338.00) | 0.43 |
| C-PR (pg/ml) | 4.36 (8.00) | 4.10 (6.00) | 5.89 (15.00) | 0.09 |
| Kt/v | 1.39 (0.37) | 1.34 (0.37) | 1.43 (0.36) | 0.28 |
| Hemoglobin (g/dL) | 10.71±1.82 | 10.81±1.92 | 10.42±1.50 | 0.28 |
| Hematocrit (%) | 32.77±5.47 | 33.30±5.53 | 31.44±5.17 | 0.09 |
| Albumin (g/dL) | 3.90±0.35 | 3.95±0.34 | 3.77±0.34 | 0.01* |
| Globulin (g/dL) | 3.00 (0.60) | 3.00 (0.60) | 3.10 (0.88) | 0.18 |
| Potassium (g/dL) | 5.40±0.73 | 5.40±0.73 | 5.39±0.72 | 0.91 |
| Sodium (g/dL) | 138.38±2.70 | 138.46±2.50 | 138.17±3.18 | 0.60 |
| Phosphorus (g/dL) | 4.83±1.50 | 5.04±1.51 | 4.31±1.40 | 0.01* |
| Calcium (mg/dL) | 8.95±0.86 | 8.96±0.89 | 8.95±0.78 | 0.96 |
| LDL (mg/dL) | 88.86±28.86 | 86.02±28.73 | 96.43±28.17 | 0.09 |
| HDL (mg/dL) | 38.76±14.91 | 37.37±12.84 | 42.34±19.05 | 0.11 |
| Triglycerides (mg/dL) | 116.00 (87.50) | 117.00 (92.00) | 154.00 (97.00) | 0.56 |
| Ferritin (ng/dL) | 418.50 (340.00) | 215.00 (380.95) | 401.00 (327.75) | 0.20 |
| Iron (µg/dL) | 54.00 (21.75) | 54.00 (27.00) | 54.50 (25.25) | 0.78 |
| Alkaline phosphatase (U/L) | 107.00 (99.25) | 97.00 (89.50) | 118.00 (91.75) | 0.19 |
| **Body composition** | | | | |
| Dry weight (kg) | 63.66±15.15 | 63.98±15.78 | 62.88±15.73 | 0.72 |
| Abdominal circum. (cm) | 81.00 (21.00) | 79.00 (20.00) | 86.00 (24.00) | 0.24 |
| BMI (Kg/m$^2$) | 22.95 (8.00) | 23.90 (8.00) | 22.40 (9.00) | 0.51 |
| Body fat (%) | 30.14±10.51 | 29.04±10.26 | 33.01±10.80 | 0.07 |
| ALM (kg) | 30.69±14.21 | 32.24±14.72 | 26.64±12.06 | 0.64 |
| **Physical test** | | | | |
| Gait speed (m/s) | 1.14±0.29 | 1.23±0.26 | 0.90±0.20 | <0.0001* |
| Low mobility (yes/no) # | 52(43)/ 68(57) | 22 (26)/ 64 (74) | 30 (88)/4(35) | >0.001* |

cardiovascular diseases [36]. This is a very sensitive measure that in older populations is considered a sixth vital sign [16] and a functional vital sign [17].

In hemodialysis patients, gait speed has been associated with frailty diagnosis [20], hospitalizations [5], and cardiovascular events [21,22]. Previous studies demonstrated an association between gait speed and mortality. [5,21,23] Nevertheless, the predictive value of gait speed remained underexplored.

In our analysis, a 1m/s increase in gait speed was associated with a 96% reduction in the risk of death. Also, patients with low mobility had a risk of death 10.4-fold higher than those who had normal gait speed. Reduced gait speed may be a marker of an underlying process. Firstly, we also caution that a reduced gait speed may reflect the functional capacity and overall health status of an individual. [16,17] Lower gait speed may indicate underlying health issues, such as

**Table 2. Univariate and multivariate Cox regression analysis of mortality. Adjusted model: For sex, age, body mass index (BMI), hemodialysis vintage, presence of diabetes, hemoglobin level, P, and albumin level. HR hazard ratio, 95% CI 95% confidence interval. *p < 0,05; # ≤ 1.13 m/s.**

| Variables | Crude Model | | Adjusted model | |
|---|---|---|---|---|
| | HR (95% CI) | p-value | HR (95% CI) | p-value |
| **Model 1** | | | | |
| Age (years) | 1.05 (1.01-1.09) | 0.01* | – | 0.71 |
| Phosphorus (g/dL) | 0.72 (0.54-0.96) | 0.03* | – | 0.83 |
| Gait Speed (meters/second) | 0.04 (0.01-0.14) | <0.0001* | 0.04 (0.01-0.15) | <0.0001* |
| **Model 2** | | | | |
| Low mobility (yes) # | 11.76 (4.13-33.44) | <0.0001* | 10.49 (3.70-29.96) | <0.0001* |

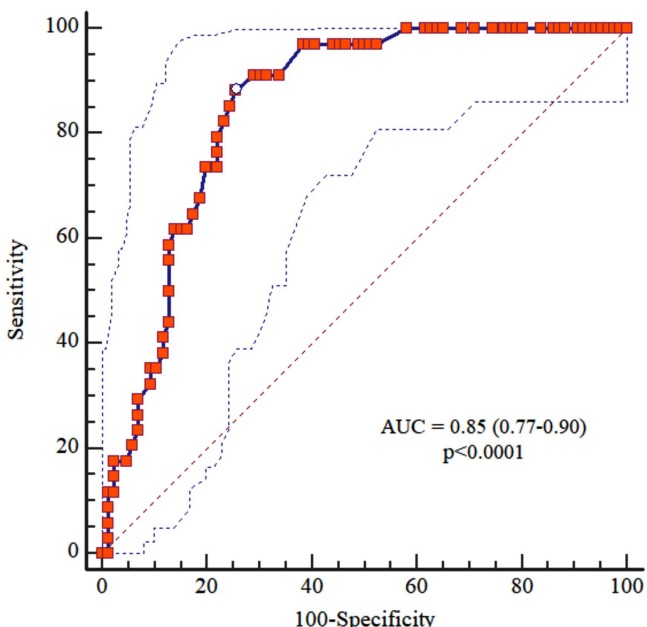

**Fig 2. ROC curves for gait speed.** ROC curves for gait speed (AUC = 0.85 [0.77-0.90]; p < 0.0001) in the mortality screening.

**Table 3. Cutoff points, AUC-curve, sensitivity, specificity, PPV, and NPV of the gait speed test for screening mortality. AUC area under the ROC curve, PPV positive predictive value, NPV negative predictive value, CI confidence interval.**

| Cutoff point | Sensitivity (95% CI) | Specificity (95% CI) | PPV (95% CI) | NPV (95% CI) |
|---|---|---|---|---|
| ≤0.61 | 11.8 (3.3-27.5) | 98.8 (93.7-100.0) | 80.0 (28.4-99.5) | 73.9 (64.9-81.7) |
| ≤1.13 | 88.2 (72.5-96.7) | 74.4 (63.9-83.2) | 57.7 (43.2-71.3) | 94.1 (85.6-98.4) |
| ≤1.28 | 100.0 (89.7-100.0) | 41.9 (31.3-53.0) | 40.5 (29.9-51.7) | 100.0 (90.3-100.0) |

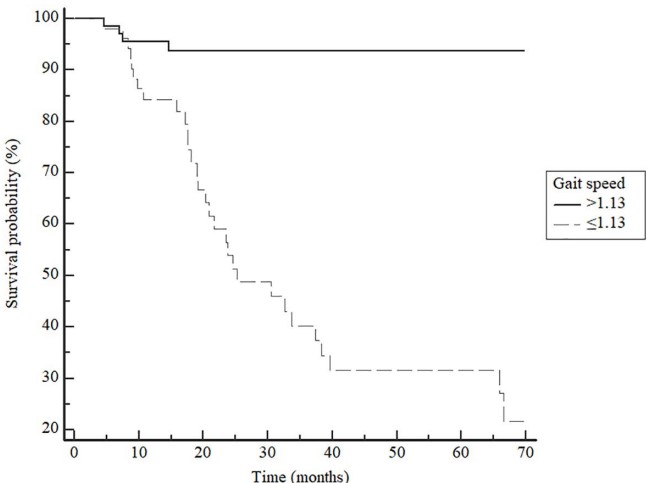

**Fig 3. Kaplan-Meier estimate of cumulative survival by gait speed categories (≤1.13 m/s or >1.13 m/s).**

muscle wasting, frailty, inflammation, or impaired cardiovascular function, a well-established predictor of poor prognosis [21,33,37,38]

The gait speed accurately predicted death in the sample. Alternative cut-off values for gait speed were proposed for use in clinical practice as appropriate. Gait speed showed low positive and high negative predictive values. The cut-off gait speed values for classifying low mobility also demonstrated a substantial negative predictive value.

The gait speed value with the best sensitivity and specificity was 1.13m/s. This cutoff point was not similar to the one found for other populations, such as patients with heart failure (≤0.8m/s) and post-cardiac surgery (0.63m/s). [39,40] In the COPD patients, a similar cutoff of 1.04 m/s was identified for impaired health status identification [41]. However, in addition to being different populations, the age mean in these studies is older. Early identification of risk in younger patients may serve as a potential strategy for reducing mortality.

Finally, in the Kaplan-Meier curve, a significant difference in survival distribution was observed among gait speed categories (Log-rank: $x^2=34.38$; $p<0.0001$), highlighting the accuracy of the gait speed test to identify mortality in this population.

In conclusion, our study supports the hypothesis that gait speed is an accurate functional test for identifying and stratifying the mortality risk. The measure of the gait speed was a protective factor, and the presence of low mobility was a risk factor for mortality. These findings highlight the prognostic value of the test and the importance of the functional evaluation in hemodialysis patients.

### Strengths and limitations

Among the strengths of the study, gait speed is a simple, fast, easy-to-use, and free-of-charge measure. Thus, the gait speed test is valuable for identifying patients with mortality hazards and who may benefit from interventions to prevent this negative outcome. The principal limitation of this study is that the cohort may not be highly representative of all hemodialysis centers. However, it is representative of dialysis centers with similar characteristics. The study was conducted in a single outpatient hemodialysis center located in a region with a low Human Development Index (HDI), which influences both patient characteristics and healthcare delivery. The center provides chronic hemodialysis predominantly within a public healthcare system, and the patient population is characterized by a wide age range, with a predominance of younger adults compared to cohorts commonly reported in high-income settings. This demographic profile reflects regional

epidemiological patterns, including earlier onset of chronic kidney disease and differences in access to preventive care. Additionally, the center operates with standard staffing levels and routinely available resources, without access to specialized rehabilitation or infrastructure, which is representative of many dialysis units in low- and middle-income regions.

## Conclusion

In conclusion, our study establishes low mobility, measured by gait speed, as an independent predictor of mortality in hemodialysis patients and its prognostic value. This finding highlights the potential use of this simple, fast, easy-to-use, and free-of-charge measure as a tool for mortality risk stratification in hemodialysis patients. Incorporating this measure into the decision-making process can aid in implementing effective preventive strategies, and a functional measure associated with survival could steer current assessment protocols and interventions to be more patient-centered, contributing to a higher standard of care. Notably, this study is the first to demonstrate the prognostic value of gait speed among hemodialysis patients in a developing country.

## Supporting information

**S1 File. Methods used for biochemical measurements.**
(PDF)

**S2 File. Data available.**
(PDF)

## Acknowledgments

The authors thank the Coordenação de Aperfeiçoamento de Pessoal de Nível Superior (CAPES), Brazil; Conselho Nacional de Desenvolvimento Científico e Tecnológico (CNPq), Brazil; Fundação de Amparo à Pesquisa do Estado de Minas Gerais (FAPEMIG), Brazil; and Hemodialysis Center of Santa Casa de Caridade de Diamantina, Minas Gerais, Brazil.

## Author contributions

**Conceptualization:** Joyce Noelly Vitor Santos, Vanessa Gomes Brandão Rodrigues, Pedro Henrique Scheidt Figueiredo, Vanessa Amaral Mendonça.

**Data curation:** Joyce Noelly Vitor Santos, Henrique Silveira Costa, Pedro Henrique Scheidt Figueiredo.

**Formal analysis:** Joyce Noelly Vitor Santos, Henrique Silveira Costa, Pedro Henrique Scheidt Figueiredo, Vanessa Amaral Mendonça.

**Funding acquisition:** Joyce Noelly Vitor Santos, Pedro Henrique Scheidt Figueiredo, Vanessa Amaral Mendonça.

**Investigation:** Joyce Noelly Vitor Santos, Tamara Cunha, Elisângela Andrade Assis Madeira, Inara Caroline Marcelino Martins.

**Methodology:** Joyce Noelly Vitor Santos, Vanessa Gomes Brandão Rodrigues, Ana Cristina Rodrigues Lacerda, Henrique Silveira Costa, Pedro Henrique Scheidt Figueiredo, Vanessa Amaral Mendonça.

**Project administration:** Joyce Noelly Vitor Santos, Elisângela Andrade Assis Madeira, Pedro Henrique Scheidt Figueiredo, Vanessa Amaral Mendonça.

**Resources:** Joyce Noelly Vitor Santos, Vanessa Gomes Brandão Rodrigues, Maria Cecília Sales Mendes Prates, Frederico Lopes Alves, Emílio Henrique Barroso Maciel, Pedro Henrique Scheidt Figueiredo, Vanessa Amaral Mendonça.

**Software:** Joyce Noelly Vitor Santos.

**Supervision:** Joyce Noelly Vitor Santos, Pedro Henrique Scheidt Figueiredo, Vanessa Amaral Mendonça.

**Validation:** Joyce Noelly Vitor Santos, Ana Cristina Rodrigues Lacerda, Henrique Silveira Costa, Pedro Henrique Scheidt Figueiredo, Vanessa Amaral Mendonça.

**Visualization:** Joyce Noelly Vitor Santos, Pedro Henrique Scheidt Figueiredo, Vanessa Amaral Mendonça.

**Writing – original draft:** Joyce Noelly Vitor Santos.

**Writing – review & editing:** Joyce Noelly Vitor Santos, Redha Taiar, Vanessa Kelly da Silva Lage, Ana Cristina Rodrigues Lacerda, Henrique Silveira Costa, Pedro Henrique Scheidt Figueiredo, Vanessa Amaral Mendonça.

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
