## [Decision Letter · Decision Letter 0]

24 Nov 2025

Dear Dr. Mendonça,

Thank you for submitting your manuscript to PLOS ONE. After careful consideration, we feel that it has merit but does not fully meet PLOS ONE’s publication criteria as it currently stands. Therefore, we invite you to submit a revised version of the manuscript that addresses the points raised during the review process.

We look forward to receiving your revised manuscript.

Kind regards,

Yuri Battaglia

Academic Editor

PLOS ONE

Journal Requirements:

“The authors thank the Coordenação de Aperfeiçoamento de Pessoal de Nível Superior (CAPES), Brazil; Conselho Nacional de Desenvolvimento Científico e Tecnológico (CNPq), Brazil; Fundação de Amparo à Pesquisa do Estado de Minas Gerais (FAPEMIG) (APQ-00709-24; APQ-01522-23), Brazil; and Hemodialysis Center of Santa Casa de Caridade de Diamantina, Minas Gerais, Brazil.”

Reviewers' comments:

Reviewer's Responses to Questions

**Comments to the Author**

1. Is the manuscript technically sound, and do the data support the conclusions?

Reviewer #1: Yes

Reviewer #2: Partly

Reviewer #3: Yes

2. Has the statistical analysis been performed appropriately and rigorously?

Reviewer #1: Yes

Reviewer #2: No

Reviewer #3: Yes

3. Have the authors made all data underlying the findings in their manuscript fully available?

Reviewer #1: Yes

Reviewer #2: No

Reviewer #3: No

4. Is the manuscript presented in an intelligible fashion and written in standard English?

Reviewer #1: Yes

Reviewer #2: Yes

Reviewer #3: Yes

Reviewer #1: Thank you for the invitation to review this paper, Dr. Bataglia, below are my comments

- Could the authors clarify whether the gait speed cutoff of 1.13 m/s was derived exclusively from their ROC analysis, or if it was also compared to previously established thresholds in similar populations?

- In the sample size calculation, the authors mention an expected event rate of 34%. Could they explain how this percentage was determined (e.g., from prior literature or preliminary data)?

- The results indicate that phosphorus was associated with mortality in univariate analysis but not in the adjusted model. Could the authors elaborate on possible reasons for this attenuation?

- Also, the paper didn't the criteria for an acceptable ROC-AUC value, consider this literature to help clarify an acceptable threshold https://doi.org/10.1080/23249935.2025.2516817 (typically 0.8 and above).

- In the causes of death reported, some were listed as “undetermined origin.” Could the authors clarify how these cases were classified and whether they might influence the robustness of the survival analysis?

- The manuscript highlights gait speed as a “free-of-charge measure.” Could the authors expand briefly on the feasibility of implementing this in dialysis centers with limited staffing or high patient loads?

- The generalizability of findings is noted as a limitation, given the single-center design. Could the authors provide more detail on the characteristics of this center (e.g., patient demographics, resources) to help readers assess comparability with other settings?

Reviewer #2: 1. Phosphorus (HR 0.72 [95% CI 0.54–0.96], p = 0.03). Can I interpret this as: For each unit increase in serum phosphorus, the risk of HD-related mortality decreases by 18%? This does not align with clinical understanding.

2. HR = 0.04 [95% CI 0.00–0.14], approaching zero, which similarly defies logic.

3. I don't understand what Table 2 is trying to convey. What does “Low mobility” refer to? And which covariates were adjusted for?

Reviewer #3: The submitted manuscript presents an extremely relevant topic in the clinical context of dialysis, presenting a variable that is easy to measure and low cost. The article has strengths regarding the sample size, analysis time, and robust statistical treatment. It is suggested that the contact information for sample recruitment be included. Considering the additional analyses of gait speed, even though an adjusted regression analysis was performed, it is considered relevant to discuss the influence of variables such as age, body composition, and significant biochemical parameters on gait speed and mobility. Although dialysis time and none of the aspects of body composition showed statistical significance, citing and discussing the possible influences on gait is fundamental (Discussion Topic).

Below are the requests for technical adjustments to the text:

• It seems that the phosphorus and albumin values are swapped in the table, as indicated in the article's discussion;

• I believe it would be interesting to briefly explain the methods for biochemical analyses;

• Adjust the table legends regarding the abbreviations.

**Do you want your identity to be public for this peer review?** For information about this choice, including consent withdrawal, please see our Privacy Policy

Reviewer #1: No

Reviewer #2: No

Reviewer #3: No

---

## [Author Response · Author response to Decision Letter 1]

15 Jan 2026

Emily Chenette, PhD

Editor

PLOS ONE

Manuscript Number: PONE-D-25-31233

Dear Editor,

I am attaching the revised version of our paper titled "The prognostic value of gait speed in hemodialysis patients: A prospective observational study” for consideration as an article in PLOS ONE. The revised sections are highlighted in yellow to emphasize the changes made.

We want to express our gratitude for the opportunity to review our document and make improvements to key points, thereby meeting the journal's requirements and addressing the reviewers’ suggestions. Formatting and quality have been adjusted in accordance with the guidelines provided in the author's guide. We have implemented all the suggested changes, and the following pages contain our point-by-point responses to the queries raised by the reviewers. We have addressed the proposed notes and provided clarification for any remaining questions.

With these revisions and responses, we believe that our work is now ready for acceptance by this esteemed journal.

Yours sincerely,

Vanessa Amaral Mendonça

The corresponding author*

REVIEWER COMMENTS:

Journal Requirements:

Answer: Thank you. Manuscript meets PLOS ONE's style requirements, including those for file naming.

Answer: This work didn’t receive funding from my institution.

Answer: The funders had no role in study design, data collection and analysis, decision to publish, or preparation of the manuscript.

Answer: The authors didn’t receive any salary from any of your funders.

Answer: This work didn’t receive any specific funding, but we must acknowledge some Brazilian agency research support. This is crucial for maintaining our line of research in Brazil. It’s essential to receive grants in our country.

“The authors thank the Coordenação de Aperfeiçoamento de Pessoal de Nível Superior (CAPES), Brazil; Conselho Nacional de Desenvolvimento Científico e Tecnológico (CNPq), Brazil; Fundação de Amparo à Pesquisa do Estado de Minas Gerais (FAPEMIG) (APQ-00709-24; APQ-01522-23), Brazil; and Hemodialysis Center of Santa Casa de Caridade de Diamantina, Minas Gerais, Brazil.”

“The authors thank the Coordenação de Aperfeiçoamento de Pessoal de Nível Superior (CAPES), Brazil; Conselho Nacional de Desenvolvimento Científico e Tecnológico (CNPq), Brazil; Fundação de Amparo à Pesquisa do Estado de Minas Gerais (FAPEMIG) (APQ-00709-24; APQ-01522-23), Brazil; and Hemodialysis Center of Santa Casa de Caridade de Diamantina, Minas Gerais, Brazil.”

Answer: We confirm that all raw data required to replicate the results of this study are provided. The data are available in a PDF file included as Supplementary Material. This file contains all data used in the analyses reported in the manuscript.

Answer: Thank you. The recommendation to cite specific previously published works was reviewed and evaluated.

REVIEWERS' COMMENTS:

Reviewer's Responses to Questions

Comments to the Author

1. Is the manuscript technically sound, and do the data support the conclusions?

Reviewer #1: Yes

Reviewer #2: Partly

Reviewer #3: Yes

2. Has the statistical analysis been performed appropriately and rigorously?

Reviewer #1: Yes

Reviewer #2: No

Reviewer #3: Yes

3. Have the authors made all data underlying the findings in their manuscript fully available?

Reviewer #1: Yes

Reviewer #2: No

Reviewer #3: No

4. Is the manuscript presented in an intelligible fashion and written in standard English?

Reviewer #1: Yes

Reviewer #2: Yes

Reviewer #3: Yes

5. Review Comments to the Author

Reviewer #1: Thank you for the invitation to review this paper, Dr. Bataglia, below are my comments

- Could the authors clarify whether the gait speed cutoff of 1.13 m/s was derived exclusively from their ROC analysis, or if it was also compared to previously established thresholds in similar populations?

Answer: Thank you for this comment. The gait speed cutoff of 1.13 m/s was derived from the ROC curve analysis performed in our sample, aiming to identify the threshold with the best balance between sensitivity and specificity for the studied outcome. Subsequently, we conducted a comparative analysis with gait speed cutoffs reported in previous studies involving populations with chronic diseases to assess the external validity of our findings. This comparison showed that the identified cutoff is slightly higher, which can be attributed to the younger age range of the population included in this study. This question is explored in the Discussion section (page 12, lines 247-253).

- In the sample size calculation, the authors mention an expected event rate of 34%. Could they explain how this percentage was determined (e.g., from prior literature or preliminary data)?

Answer: Thank you for this comment. An initial literature review was conducted to identify mortality rates reported in similar populations. Subsequently, a pilot study was performed, and the observed mortality rate of 34% was found to be consistent with values previously described in the literature.

References:

1. de Jager DJ, Grootendorst DC, Jager KJ, et al. Cardiovascular and Noncardiovascular Mortality Among Patients Starting Dialysis. JAMA. 2009;302(16):1782–1789. doi:10.1001/jama.2009.1488

2. Barra, A.B.L., Roque-da-Silva, A.P., Canziani, M.E.F. et al. Characteristics and predictors of mortality on haemodialysis in Brazil: a cohort of 5,081 incident patients. BMC Nephrol 23, 77 (2022). https://doi.org/10.1186/s12882-022-02705-x

- The results indicate that phosphorus was associated with mortality in univariate analysis but not in the adjusted model. Could the authors elaborate on possible reasons for this attenuation?

Answer: Thank you for this comment. Cox proportional hazards regression is a time-to-event analysis that estimates the association between covariates and the risk of the studied outcome, allowing the identification of independent predictors when multiple variables are included in the model. In the univariate analysis, serum phosphorus was significantly associated with mortality when evaluated in isolation. However, this association was attenuated in the multivariate model after adjustment for functional performance. This is a mathematical attenuation, and it likely occurred because, when analyzed together, functional performance captures a broader and more integrated representation of the patients' clinical and physiological status, thereby overlapping the prognostic information provided by serum phosphorus alone.

- Also, the paper didn't the criteria for an acceptable ROC-AUC value, consider this literature to help clarify an acceptable threshold https://doi.org/10.1080/23249935.2025.2516817 (typically 0.8 and above).

Answer: Thank you for this comment. In this study, an AUC greater than 0.7 was considered acceptable, while an AUC greater than 0.8 was considered excellent for the proposed cutoffs, based on the literature (Hajian-Tilaki, 2013). The AUC to identify mortality hazards in hemodialysis patients using gait speed was 0.85 (0.77-0.90), p < 0.0001, indicating an excellent result. We thank the reviewer for the suggestion. The recommended reference has been incorporated into the Methods and References sections (Reference 30).

Reference: Hajian-Tilaki K. Receiver Operating Characteristic (ROC) Curve Analysis for Medical Diagnostic Test Evaluation. Caspian J Intern Med. 2013;4(2):627–35.

- In the causes of death reported, some were listed as “undetermined origin.” Could the authors clarify how these cases were classified and whether they might influence the robustness of the survival analysis?

Answer: Thank you for this comment. Deaths classified as “undetermined origin” occurred in cases in which the patient died within their municipality and detailed information on the cause of death was not available in the medical records or official registries at the time of data collection. Importantly, these cases were classified as death events in the survival analysis regardless of the specific cause. As the primary outcome of this study was all-cause mortality, the absence of cause-specific information does not compromise the robustness of the Cox regression models. In addition, deaths due to external causes, such as accidents, trauma, or other non-disease-related events, were neither reported nor observed in this cohort.

- The manuscript highlights gait speed as a “free-of-charge measure.” Could the authors expand briefly on the feasibility of implementing this in dialysis centers with limited staffing or high patient loads?

Answer: Integrating the gait speed test into routine dialysis assessments is a practical and cost-effective strategy to enhance patient monitoring, making it particularly suitable for dialysis centers with limited staffing or high patient volumes. The test requires minimal and straightforward training, as the procedure involves only a standardized environment (an 8-meter corridor) and simple instructions (Cruz-Jentoft et al., 2019). It is quick to administer and can be scheduled at regular intervals, such as monthly or quarterly, without interfering with the dialysis routine. This integration enables longitudinal follow-up of functional capacity, providing prognostic and clinically relevant information with minimal burden on healthcare resources.

Reference: Cruz-Jentoft AJ, Bahat G, Bauer J, Boirie Y, Bruyère O, Cederholm T, et al. Sarcopenia: revised European consensus on definition and diagnosis. Age Ageing. 2019 Jan 1;48(1):16–31.

- The generalizability of findings is noted as a limitation, given the single-center design. Could the authors provide more detail on the characteristics of this center (e.g., patient demographics, resources) to help readers assess comparability with other settings?

Answer: Thank you for highlighting this important point regarding generalizability. The study was conducted in a single outpatient hemodialysis center located in a region with a low Human Development Index (HDI), which influences both patient characteristics and healthcare delivery. The center provides chronic hemodialysis predominantly within a public healthcare system, and the patient population is characterized by a wide age range, with a predominance of younger adults compared to cohorts commonly reported in high-income settings. This demograp

---

## [Decision Letter · Decision Letter 1]

9 Feb 2026

The prognostic value of gait speed in hemodialysis patients: A prospective observational study

PONE-D-25-31233R1

Dear Dr. Mendonça,

We’re pleased to inform you that your manuscript has been judged scientifically suitable for publication and will be formally accepted for publication once it meets all outstanding technical requirements.

Kind regards,

Yuri Battaglia

Academic Editor

PLOS One

---

## [Editor Report · Acceptance letter]

PONE-D-25-31233R1

PLOS One

Dear Dr. Mendonça,

I'm pleased to inform you that your manuscript has been deemed suitable for publication in PLOS One. Congratulations! Your manuscript is now being handed over to our production team.

Kind regards,

on behalf of

Prof. Yuri Battaglia

Academic Editor

PLOS One